# The Ball Response on the Beech Parquet Floors Used for Basketball Halls

Aurel Lunguleasa *, Cosmin Spirchez, Loredana Radulescu and Minerva Turcas Diaconu

Department of Wood Processing and Design of Wood Products, Faculty of Furniture Design and Wood Engineering, Transilvania University of Brasov, 1 Universitatii 1, 500068 Brasov, Romania; cosmin.spirchez@unitbv.ro (C.S.); loredana@unitbv.ro (L.R.); olimpia.turcas@unitbv.ro (M.T.D.)
* Correspondence: lunga@unitbv.ro

**Abstract:** In recent years, with the occurrence of standards in the field, the realization of parquet floors in basketball halls acquires new values that take into account, among other factors, the response of the ball to touching the floor. That is why the paper aims to test four beech parquet floor structures in order to find optimal solutions for these sports activities. Each structure with an area of 1 m × 1 m includes beech parquet with friezes glued together with vinyl adhesive, fixed on a support of longitudinal spruce slats 20 mm thick in the case of structure A, fixed on a spruce frame in the case of structure B, fixed on a spruce frame and beech taggers in the case of structure C, and fixed on a spruce frame and rubber taggers in the case of structure D. The results of laboratory tests showed the clear advantages of type B structures, of type C structures with a large number of beech shock pads, and D-type structures with a small number of rubber shock pads. All tests were based on the antagonism of the elasticity and rigidity properties of the beech wood. Through methodology and experiment, the research contributes to the construction of beech parquet floors used in basketball halls, in terms of the ball's response to touching the floor.

**Keywords:** ball response; beech wood; parquet flooring; wooden slipper





## 1. Introduction

The flooring and parquet industry is a topical and constantly developing industry, which is in line with the market of used wood materials as well as new construction and installation systems. The more sophisticated the parquet and floor technology becomes, the more the surface of the basketball halls is taken into account, which requires a series of additional conditions related to the protection and performance of the player but also to the reaction of the ball to touching the floor. From the beginning, the floor of the basketball halls had a layer of resistance concrete on which there was put a layer of wooden rulers as an elastic layer and finally the parquet or the floor itself as an element of resistance, elasticity, non-slipping, and decoration. Knapic et al. [1] analyzed the properties of solid wood plates from cork oak (*Quercus suber* L.), highlighting the fact that in addition to high elasticity, these panels have other properties such as dimensional stability and wear resistance as good as other oak species (*Quercus robur*, *Quercus petraea*, *Quercus pubescens*). Bergès et al. [2] highlighted other qualities of cork oak, such as the optimal density for the construction of sports hall floors. Lamason and Gong [3] went in the same direction as the previous authors, namely the use of soft and light wood species such as poplar to obtain light and very elastic floors. Hacibektasoglu et al. [4] analyzed some of the mechanical properties of reconstituted panels made of solid beech wood slats (*Fagus sylvatica* L.) that were heat treated at high temperatures of 200 °C for 2.5 h and were compared with untreated panels. Deteix et al. [5] proposed a simple solution to lighten the structure of the wood floor by milling on the bottom of the friezes, by this method obtaining a substantial improvement of the floor elasticity. The spearhead in the field of sports floors is bionic flooring, obtained by the international company URSA through high ethnic level

technologies that optimize the elasticity of the structure with wear and resistance to slip, heat, thermal comfort, and special aesthetic appearance. Kolitzus [6] highlighted the role of sports floor testing facilities and proposed a series of tests that are currently found in European standards EN 14 808 and EN 14809. Other wood species such as sugar palm (*Arengapinnata*) [7] or *Eucalyptus globulus* [8] were tested. Basketball floors must meet the requirements of standard SR EN 12235:2014) [9]. According to this European Standard, a basketball that falls freely from a height of 1800 mm must have a relative rebound height of at least 90% of the height measured on concrete floor (this floor is taken as a comparison element). There have been various authors who have studied the reaction of the ball falling from a certain height, having as the variables different hard surfaces [10–12] and different pressures of the ball [13].

The game of basketball was created by Professor James Naismith of Springfield College in the United States, Massachusetts, in 1891, for students, in the desire to make the game of ball more attractive both in gyms and outdoors. The principles of the game at that time are still relevant today. Basketball is one of the most popular sports in the world, which emphasizes the spirit of competition and team and contributes to the harmonious development of the human body [14].

The technical performances of the sports floors (including the basketball ones) are standardized in order to protect the athlete but also to create an environment adequate to the sports performance of the player. First of all, the European standard EN 14904:2007 stipulates that sports floors have a shock absorption produced at a fall of at least 25%, a coefficient of friction to ensure the adhesion of 0.8–0.11, a vertical deformation due to the elasticity of maximum 5 mm, an abrasion resistance of at least 1 g per 1000 wear cycles, and a flatness of maximum 6 mm/3 m. Another standard in this field is EN 12235:2013 [9], which presents a method for determining the height of the ball ricochet when it falls freely from a height of $1.8 \pm 0.01$ m. The return height of the basketball on a concrete surface must be $1.050$ m $\pm 0.025$ m, and the ball must be approved by the International Basketball Federation FIBA [15,16]. The American standard ASTM F 2117-10 [17] has created a method of comparative testing of playing surfaces with reference surfaces. It proposes a sound analysis for determining the falling and lifting times and an area of the sample to be analyzed of 2 m × 2 m for surfaces made of elastic materials and 1 m × 1 m for surfaces that have punctually improved elasticity. The test must be done on five different points located in the central area of the analyzed surface.

Acura et al. [18] performed dynamic hardness tests for wooden floors of *Eucalyptus globulus* and *Eucalyptus granilis*. To compare the properties of the floors, a dynamic impact hardness test was developed based on the impact of steel balls, with several diameters and the number of falls. The dynamic tests were made with three steel balls (diameter 30, 40, 50 mm) and were performed from five different fall heights (0.60; 0.75; 0.90; 1.05; 1.20 mm). Sydor et al. [19] performed deformation tests on flooring materials (plywood, beech, pine, iroko, oak, maple, red oak, HDF laminate). The permanent (elastic) and temporary imprint of the deformations were determined. Measurements based on depth of identification have been found to be better when floors are made of hard lignocellulosic material. Schwarzkopf [20] presented the importance of thermo-hydro-mechanical treatments, which can improve certain properties of low-density wood species. These treatments densify the wood by softening the cell walls using heat, pressure, and humidity. Thermo-hydro-mechanical tests have increased the scratch resistance, density, and hardness of poplar wood used for flooring. Kureli et al. [21] in their study showed the importance of determining the effect of different wooden floor structures and surface characteristics on water supply, contraction, and swelling under different conditions of relative humidity and water retention. Nine types of wood flooring samples were tested (solid beech wood flooring covered with polyurethane varnish, four types of wood flooring with different core structures, four types of laminate flooring with different core structures). The results showed that the lowest values of water absorption for 2 h and 24 h were recorded in laminated wood floors with high-density fibers and medium-density fibers. The lowest

coefficient of swelling in thickness was recorded for laminate wood laminate flooring during exposure to high relative humidity. The lowest width coefficient was recorded for laminate flooring.

Colino et al. [12] analyzed the response of sports surfaces in terms of mechanical properties by using two types of artificial players, which were called artificial athlete and advanced artificial athlete. The shock absorption and the vertical deformation of the sports surface were determined by the action of 50 attempts. Bjelica et al. [13] studied the reaction of the ball to touching the playing surface depending on the pressure in the ball. A standard pressure and two other pressures within ±5% were used. The conclusion of the study was that the reaction of the ball will increase by increasing its internal pressure. De Oliveira et al. [22] in their study described the use of viscoelastic compression to investigate the increase in strength and stiffness of chemically modified wood. Brinell density and hardness increased by 84% and 209%, respectively. FTIR analysis showed that wood polymers present in chemically modified wood are susceptible to degradation in the presence of heat. Huang et al. [23] in their study presented the effects of process parameters (adhesive type, pressing time, pressure) on the bonding performance of beech wood floors with EPI adhesives. The results showed that the shear strength and the aging test are influenced by the type of adhesive, temperature, and pressing time. Kutnar et al. [24] in their study presented the importance of thermo-hydro-mechanical treatments. Heat treatment was applied to the beech wood with steam (pressure 620 kPa) and heat (T = 170 °C), followed by densification and temperature at 200 °C, in a hot press which was then cooled while under pressure. Czajkowski et al. [25] in their study presented the importance of beech wood properties. Beech wood is a versatile material used in construction. Liu et al. [26] in their study present a carbon fiber composite plywood. The mechanical properties, thermal insulation, and sound insulation of composite plywood were evaluated. The results showed that the composite plywood had a density of less than 0.7 g/cm$^3$. The absorption and swelling in thickness of the composite plywood were significantly reduced after the surface was hardened with carbon fiber fabric. Turcas Diaconu et al. [27] in their study investigated the performance of eight different beech wood floor structures using floor strips with the same length and width of 20 mm and thickness of 15 mm. The tests were performed according to SR EN 12235 [9]. Sepliarsky et al. [8] highlighted the importance of strength for the two types of floors made of *Eucalyptus globulus* and oak (*Quercus robur*), using the impact test according to ASTM D1037-99. Ayata et al. [28] in their study performed tests on different thermal aging regimes (30 °C for 30 days, 60 °C for 60 days, 90 °C for 90 days) for beech, maple, and oak parquet structures. Color and gloss were determined before and after the thermal aging process. Turcas (Diaconu) et al. [14,27] presented the results of experiments performed on beech floor structures for gyms. These structures are compared with the limits imposed by SR EN 1223:2014 [9] for determining the vertical behavior of the ball (Figure 1).

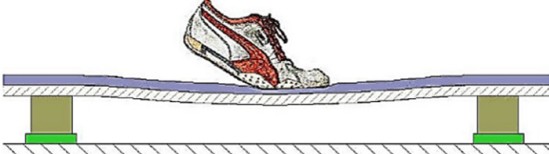

**Figure 1.** The reaction of an elastic floor to the force exerted by the basketball player.

The main elements that emerge from the study of references in the field are the following:

- So far, various composite materials have been used, with no heat or chemical treatments;
- Different wood species have been used, the best results being obtained with hard species that are dense, such as oak;
- Few studies have used the reaction of the ball to touching the floor; the vast majority use artificial athletes or mechanical properties such as Brinell hardness.

As a main conclusion of the above bibliographic study, it can be said that wood has a series of characteristics such as elasticity, good thermal and sound insulation, but also special color and aesthetics, which make it suitable for use in the floors of basketball halls. That is why the authors' attention in this field was directed toward finding optimal parquet structures made of beech wood (*Fagus sylvatica* L.) to be tested in the laboratory regarding the action of ricochet of the ball in the conditions of European and American Standards in the field. In this sense, it was proposed to use the surface of beech friezes and the laying frame of the frieze of carpet made of softwood slats with or without wooden and rubber taggers.

## 2. Materials and Methods

### 2.1. Determinations Regarding the Elasticity of Beech Wood

The wood material used in the work was purchased from the Forex company (Ghimbav, Brasov, Romania) in the form of steamed and dried beech timber with thicknesses of 25 and 50 mm. Through specific processing, specimens were obtained for combinations of 4 lamellae thicknesses (15, 20, 25, and 30 mm), 4 lamellae widths (30, 40, 50, and 60 mm) and 4 different lengths (300, 350, 400, and 450 mm), for a total of 64 specimens. The conditioning of the specimens was performed at a temperature of $20 \pm 2\,^{\circ}\text{C}$ and the relative humidity of 65 + 5% for 24 h to obtain a humidity of approximately 12%. Respecting the working principles and the devices related to the determination of the elastic modulus on the universal testing machine WE 10A (Shanghai, China), the force necessary to obtain deformations of 2, 4, 6, and 8 mm was determined. In addition, the test was continued until the specimen was broken in order to determine the modulus of elasticity. The distance between the supports of the bending device was adjusted according to the thickness of the specimen at $l_1 = 250$, $l_2 = 300$, $l_3 = 350$, and $l_4 = 400$ mm, corresponding to the 4 lengths of specimens analyzed.

After obtaining the forces corresponding to the 4 deformations that were taken into account, the lifting of the specimen with the supports was continued until the bending deformation of the specimen reached the maximum value until the rupture of the specimen, thus finding the maximum breaking force, at which time the force on the machine's dial was read. The relations for determining the modulus of elasticity at static bending were the following:

$$E_i = (l^3 \times (P_2 - P_1)) \div (4 \times b \times g^3(f_2 - f_1))\ [\text{N}/\text{mm}^2] \tag{1}$$

where l—distance between supports, in mm; $P_2 - P_1$—the first force being 10% of the breaking force and the second about 40% of the breaking load, in N; b—width of the specimen, in mm; g—specimen thickness, in mm; $f_2 - f_1$—the deformation difference of the specimen, corresponding to the two above forces, in mm.

### 2.2. Determinations Regarding the Choice of the Optimal Parquet Structure as a Reaction of the Basketball

As materials, steamed and dried beech timber were used, as specified in the previous paragraph for making parquet lamellae, softwood (spruce) timber with a thickness of 20 mm was used to obtain parquet support, and wooden and rubber taggers were used to obtain shock pads with high elasticity. The used rubber was purchased from the company in the form of a carpet with a thickness of 20 mm. The material provided by SC CONTRAX SRL (Brasov, Romania) was used to make the rubber shock pads. The rubber taggers were made of SBR (styrene–butadiene) for general use, with good physical and mechanical properties, and they were resistant to water and air. The characteristics of the material are: black color, Shore A 65 hardness, resistance 40 daN/cm$^2$, elongation at break at least 150%, thickness 10 mm, density 1450 kg/m$^3$, and working temperature $(-30 + 70)\,^{\circ}\text{C}$.

Parquet friezes were made in a mechanical processing workshop, which were glued with JOWACOLL® 103.05 type adhesive (Detmold, Germany), a polyvinyl dispersion characteristic of gluing solid wood lamellar parquet, forming 4 different types of structures

with dimensions of 1.0 m × 1.0 m. Each parquet structure was glued on softwood lamellae or the related frame, with or without elastic taggers made of beech wood or synthetic rubber. A grid was made on each panel as in Figure 2, thus identifying 5 distinct points in the central area of the panel in an area of 250 mm × 250 mm, where to perform tests on the reaction height of the basketball when falling on the support.

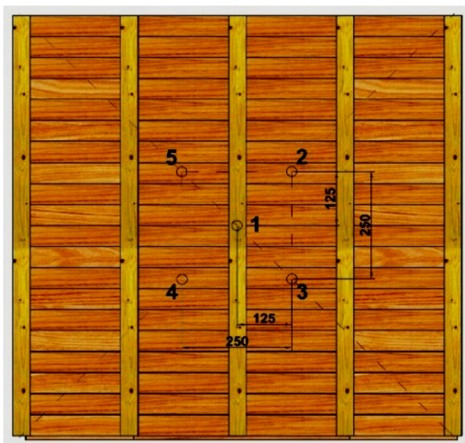

**Figure 2.** The five points of the basketball falling on the tested panel.

The calculation relation of the basketball ball reaction (R) when touching the floor was the following:

$$R = (H_s \div H_c) \times 100 \ [\%] \tag{2}$$

where $H_s$ is the return height of the ball on the sports surface, in m; $H_c$ is the height of return of the ball on the concrete surface (concrete) with self-leveling screed in mm.

The energy absorption of the tested panel was calculated with the following relation:

$$Ea = (H_c - H_s) \div 1.8 \times 100 \ [\%]. \tag{3}$$

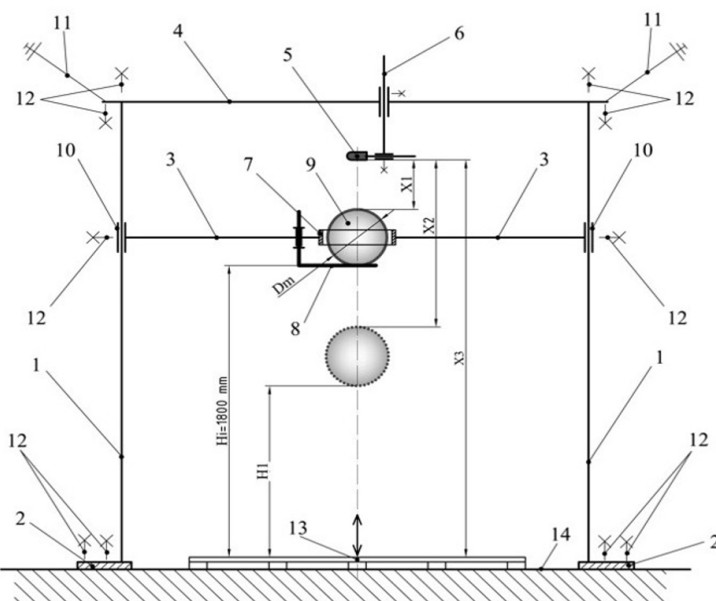

**Figure 3.** Sketch of the basketball ball reaction test stand: 1—support poles; 2—base plate for column; 3—metal ring supporting beam; 4—grid for the ultrasound device; 5—ultrasound device; 6—ultrasonic device fixing rod; 7—metal flat ring for the ball; 8—ball height diameter; 9—basketball; 10—lower beam clamping element; 11—ceiling fixing cable; 12—fixing screws; 13—carpet of beech wood friezes; 14—concrete support for tested panel.

Figure 3 shows the scheme of the experimental installation used in the basketball reaction tests. The height of the ball falling from the device was 1.8 m, and the ball release device was designed so that the ball did not roll during the fall.

Figure 4 shows the back of the 4 types of test panels marked with A, B, C, and D, respectively parquet with support-only slats of spruce timber planed with a thickness of 20 mm in type A structure (Figure 4a), with parquet arranged on the frame support from spruce timber in the case of type B structure (Figure 4b), with parquet supported by a spruce frame and beech shock supports in the case of structure C (Figure 4c) and from beech parquet with frame support from spruce timber and rubber shock pads in the case of structure D (Figure 4d).

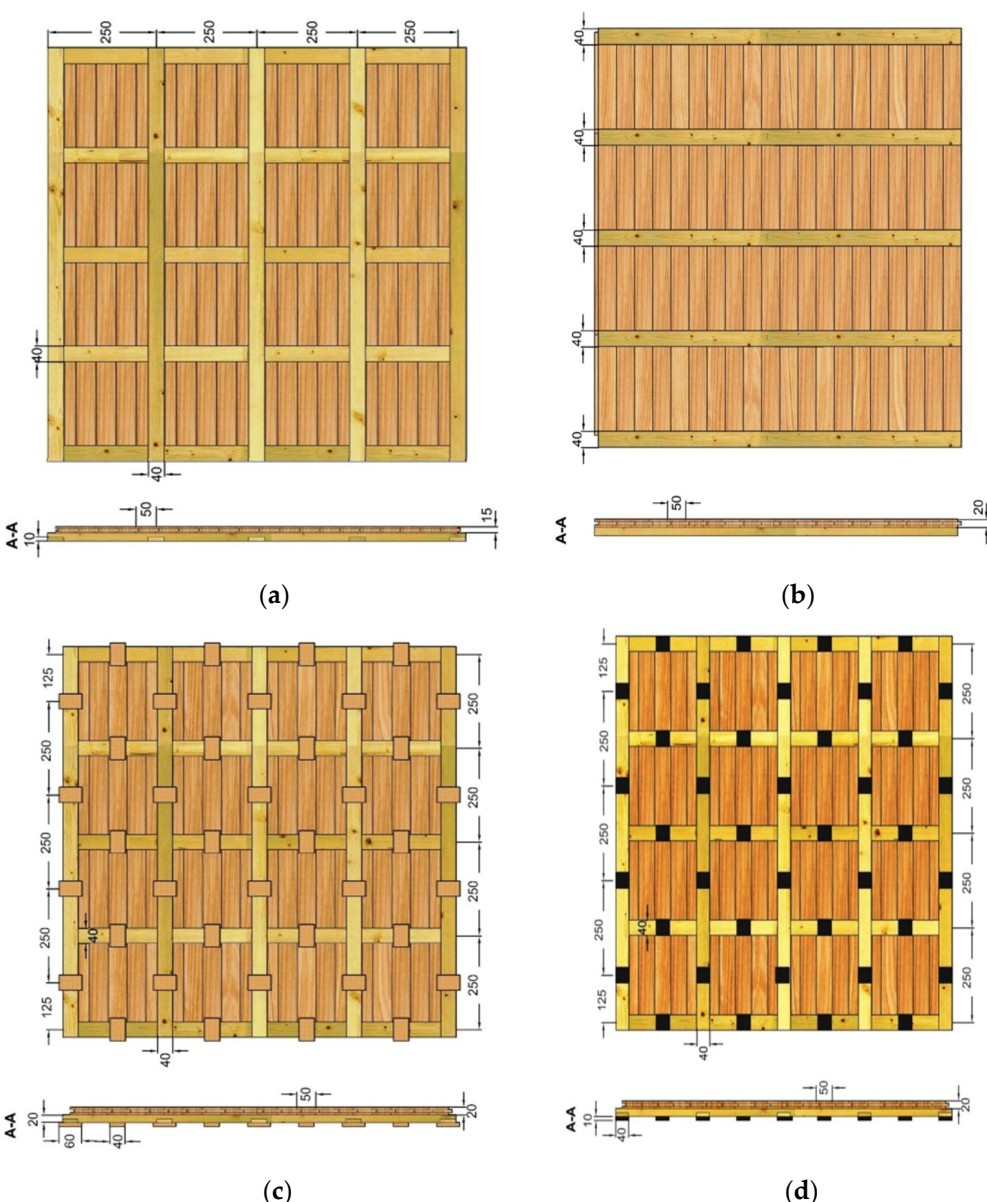

**Figure 4.** Floor structures (back) for basketball halls (A–A cross-section): (**a**) with parallel softwood slats; (**b**) with grid (in the form of a chessboard) made of softwood slats; (**c**) made of softwood slats and beech shock pads; (**d**) made of softwood grid in the form of a frame and rubber shock pads.

Each of the above structures had several variants; respectively, structure A had two types with 3 or 5 slats of support, structure B consisted of 5 side members and 5 girders made of spruce timber with parquet thickness of 15, 20, and 25 mm, structure C had 13, 25, and 40 pieces of beech shock pads, and structure D had 13, 25, and 40 pieces of synthetic rubber shock pads (Table 1). The basketball used during the laboratory tests was Molten 7, which is an official game ball for competitions organized under the auspices of the International Basketball Federation (source), with a mass of 605 g, a circumference of 754 mm, a standard inflation pressure of 0.6–0.7 bar, and a diameter of 240 mm.

**Table 1.** All structures of experimental test (lf.—long friezes; sl.—sleepers).

| Structure | Thick = 15 mm | | | Thick = 20 mm | | | Thick = 25 mm | | |
|---|---|---|---|---|---|---|---|---|---|
| A | 3 lf. | | 5 lf. | 3 lf. | | 5 lf. | 3lf. | | 5 lf. |
| B | | 5 lf. | | | 5 lf. | | | 5 lf. | |
| C | 13 sl. | 25 sl. | 40 sl. | 13 sl. | 25 sl. | 40 sl. | 13 sl. | 25 sl. | 40 sl. |
| D | 13 sl. | 25 sl. | 40 sl. | 13 sl. | 25 sl. | 40 sl. | 13 sl. | 25 sl. | 40 sl. |

The classic basketball ball reaction tester consists of several successive layers, namely a 130 mm reinforced concrete resistance floor, a 30 mm mortar layer, a 10 mm mosaic concrete coating, and a screed self-leveling slab of 2–3 mm. In order to obtain the reaction of the basketball when falling on the concrete support, the average of 5 values of the return height on its surface was registered. The tests were performed on each type of panel, after which a comparative analysis of the ball reaction on each surface compared to the concrete surface was performed, finally obtaining the optimal structures, which exceed the minimum reference values.

From a statistical point of view, the main statistical parameters of trend and spread were calculated, respectively the arithmetic mean and the standard deviation. Where necessary, the trend curve was found that best approximates the evolution of the experiment values, for a Pearson $R^2$ determination coefficient greater than 0.9.

## 3. Results and Discussion

The results obtained referred to the two aspects taken into account, namely the choice of friezes in order to create structures with high elasticity and strength for floor stability and the type of structure to ensure a good response of the basketball touching the parquet surface.

### 3.1. Choosing the Dimensions of the Friezes Depending on Their Elasticity

Based on the test plan and the methodology adopted, when establishing the force required to achieve a certain deformation that increased by 2 mm ratio, we observed first of all a proportional increase of the force with the increase of the deformation, after a linear curve with a very good Pearson coefficient, with the value $R^2$ = 0.9991 (Figure 5). A percentage increase of the force was also observed in the range from 2 to 4 mm deformation for the lamella of 30 mm × 30 mm × 650 mm by 60%, which is kept on the following ranges from 4 to 6 mm and from 6 to 8 mm (Figure 6). This uniform growth is given by the uniformity of the structure of the beech wood (it is a species of hardwood with uniformed spread pores) and the fact that was steamed but also by the fact that all the defects existing in the pieces of steamed timber were eliminated when the friezes were processed. The modulus of elasticity obtained during the tests, by calculation using relation (3), had an average value of 10,992 N/mm$^2$ with a standard deviation of ±2743 N/mm$^2$. All the average values obtained for different widths and thicknesses of specimens but also distances between supports are visible in the graph in Figures 7 and 8. It is obvious that with the increase of the specimen thickness, the modulus of elasticity decreases significantly from about 15,000 to 8000 N/mm$^2$, which means a percentage decrease of 46.6%. Knapic et al. [1] found that the oak species has a good elasticity, without detailing the fact that this species is very expensive, and it is recommended to use it for aesthetic veneers and good quality timber. By approaching the four straight lines and by their intersection, it is highlighted that the width of the specimen has very little influence on the modulus of elasticity.

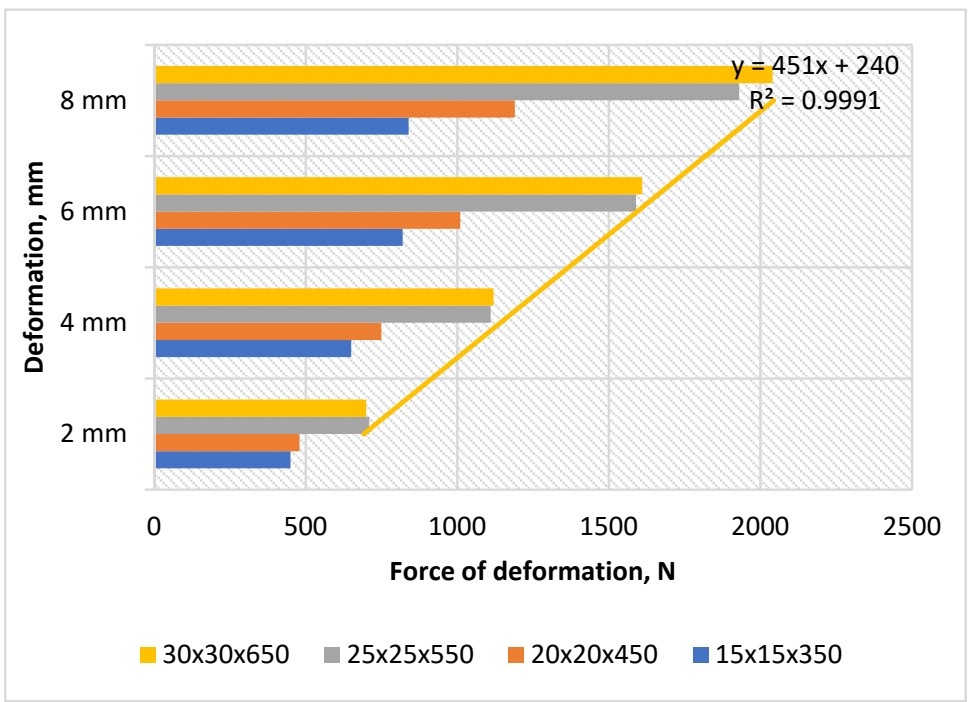

**Figure 5.** Evolution of the force depending on the adopted deformation of 2, 4, 6, and 8 mm.

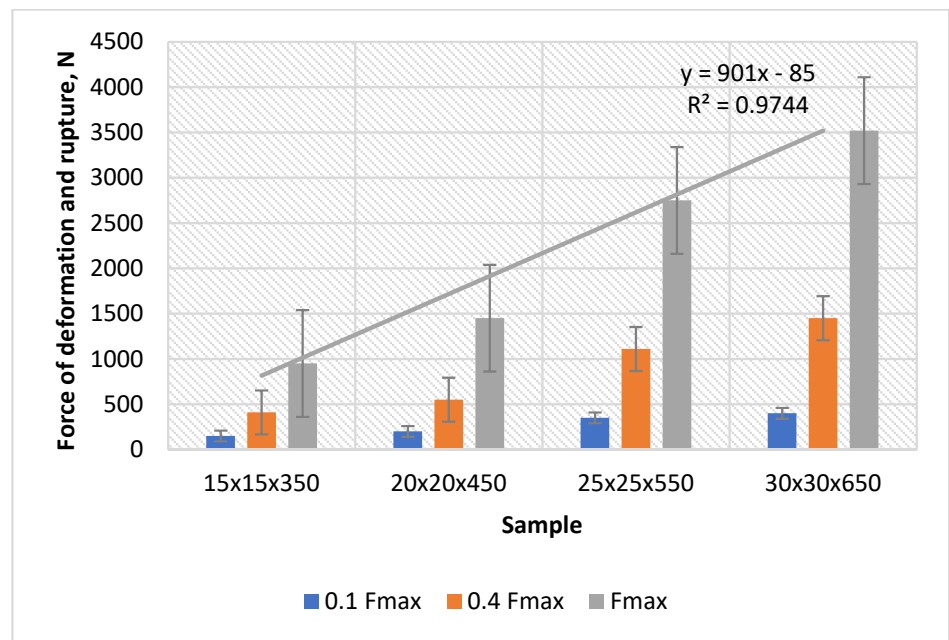

**Figure 6.** The evolution of the deformation forces until the breaking of the specimen.

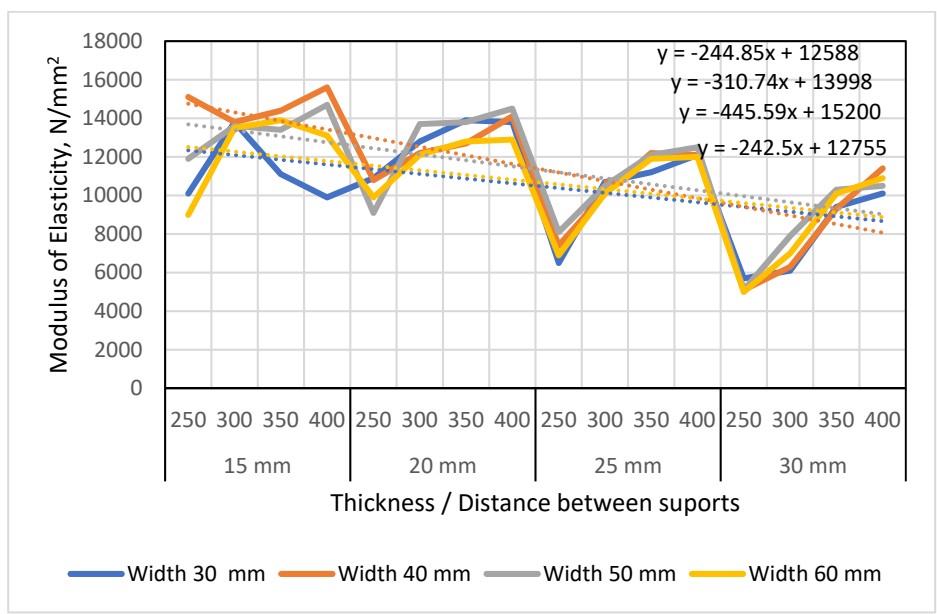

**Figure 7.** Influence of the width and thickness of the specimen on the modulus of elasticity.

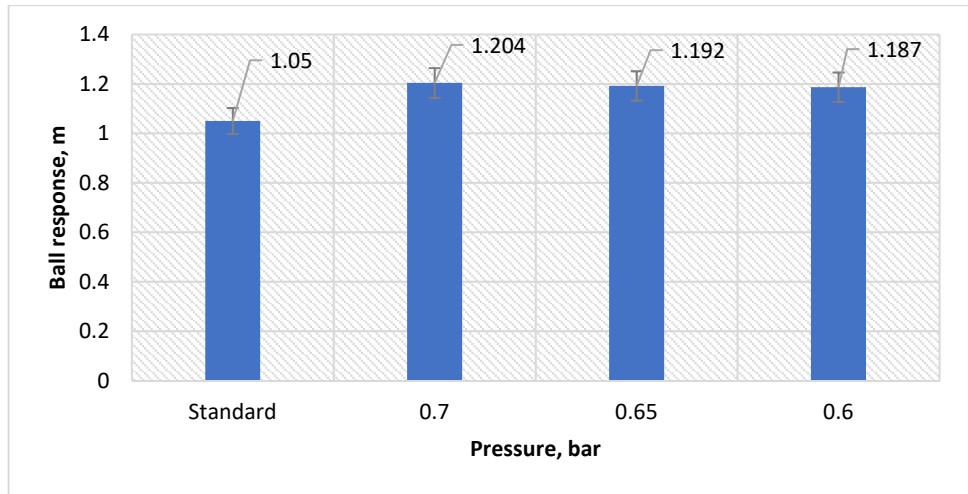

**Figure 8.** Values for the maximum bounce of the ball on the concrete surface at three different pressures in the ball.

Since the modulus of elasticity is maximum for thicknesses of 15 and 20 mm, for technological reasons (easy processing and more rigidity), the thickness of 20 mm was chosen. Based on the results obtained after testing deformation for elasticity by bending, the optimal dimensions used to make the test panels were established, from this point of view, the friezes with dimensions of 500 mm × 50 mm were found as optimal.

### 3.2. Choosing the Optimal Parquet Structure Depending on the Ricochet of the Ball

The four types of floor structures considered had different results. First of all, the influence of the inflation pressure of the basketball on the response when it touches the reinforced concrete surface with an equalizing slab was analyzed. Figure 8 shows the maximum bounce of the inflated ball with three different pressures compared to the average of the five measurements with the value provided in the standard.

It can be seen from Figure 8 that regardless of the pressure in the ball, measured with the pressure pencil, all ricochets exceed the standardized value, and the differences are very small; there is a difference of 0.017 m between the maximum and minimum value,

respectively an increase of 1.4%. As expected, the decrease in pressure in the ball led to a decrease in its reaction to touching the floor. Following the research, it was decided to choose the pressure of 0.7 bar, which falls within the specifications of the basketball ball manufacturer but also offers a maximum response of the ball. According to this pressure from the ball, the five measurements performed on the standard concrete floor resulted in an average bounce value of 1.204 m. This value will be retain as the value to which all the measurements performed on the wooden beech floor structures will be compared. Considering that the parquet floors must have a reaction of the ball according to the SR EN 12235:2014) of 90% of the value of the ricochet on the concrete floor, we will obtain a minimum admissible value of 1.083 m. In addition, as a comparative value, but of the absorption this time, the floors for basketball halls must have an energy absorption of a maximum of 6.69%, which is a value resulting from the calculation using relation (3) by using the two previous values of the reaction height of the basketball. Any value obtained during the experiments that will not fall within the limits specified above makes the panel structure inadequate.

The values of the ricochet of the ball on the type A panels, with longitudinal slats from softwood, are presented in Figure 9. It is observed that the average response of the ball in case of the structure with three longitudinal slats with the value 0.89 m is lower than in the case of the parquet support of five longitudinal slats made of spruce wood (with an average value of 0.923 m). This means that the structure with five longitudinal slats on the tested surface of 1 m × 1 m is more rigid and offers a better response of the basketball to touching the floor. However, in total, both values are below the minimum reference value (ricochet height of 1.083 m), which is why these floor structures are not recommended for use.

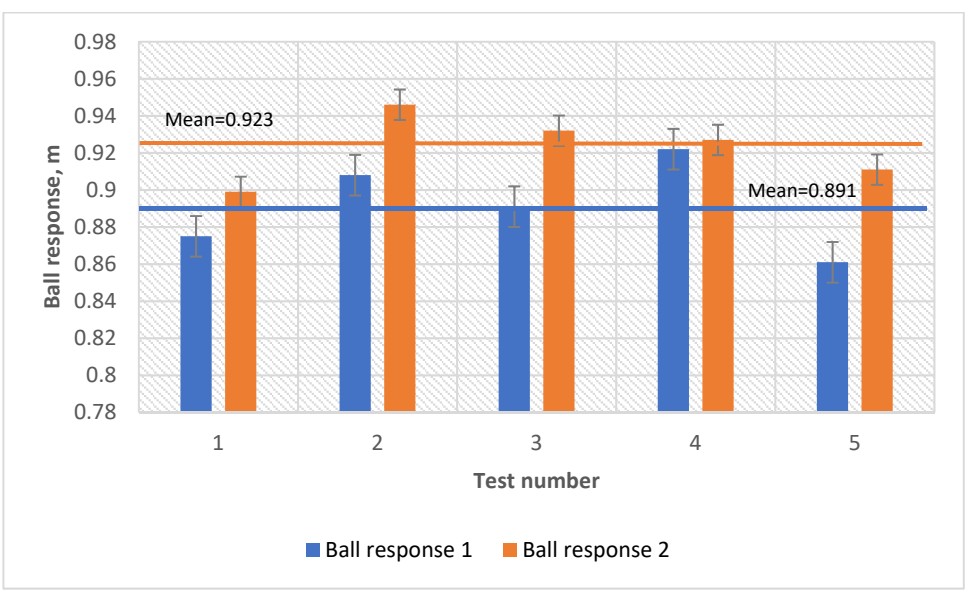

**Figure 9.** Ball response in the case of the type A basket surface with three long strips (Ball response 1) and five longitudinale strips (Ball response 2).

The energy absorption calculated using relation (3) of the type A panel supported on three longitudinal slats was 17.36%, and in the case of the panel supported on five longitudinal slats, the energy absorption of the panel was 15.62%, both exceeding the maximum value of reference of 6.69%. It can be concluded that these types of panel structures are not recommended for the surface of a basketball hall from the perspective of energy absorption. Appropriate values were found by other authors in their works [10,14].

Regarding the type B structure, made of beech parquet supported by five stringers and five girders in the form of a frame, they were differentiated only according to the thickness of the parquet friezes; respectively, there were three frieze thicknesses of 15, 20, and 25 mm (Figure 10).

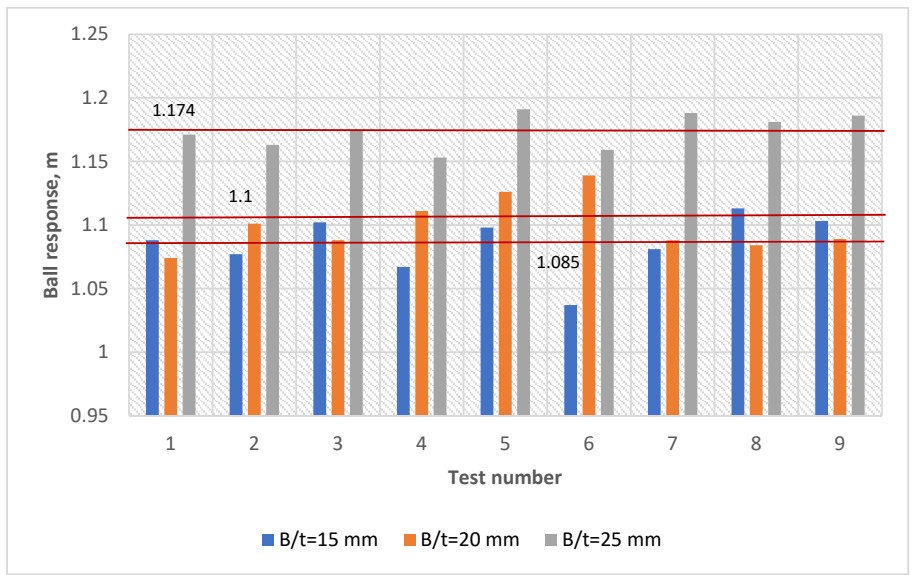

**Figure 10.** Basketball ball response on type B panels and different thicknesses of parquet friezes.

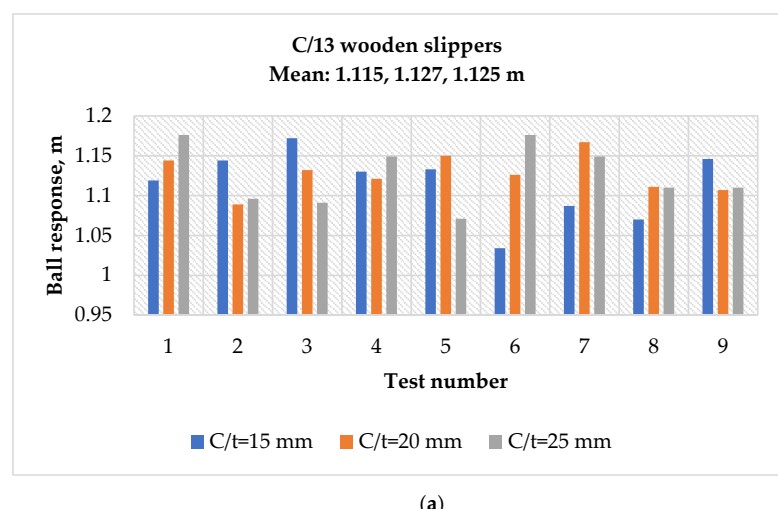

(**a**)

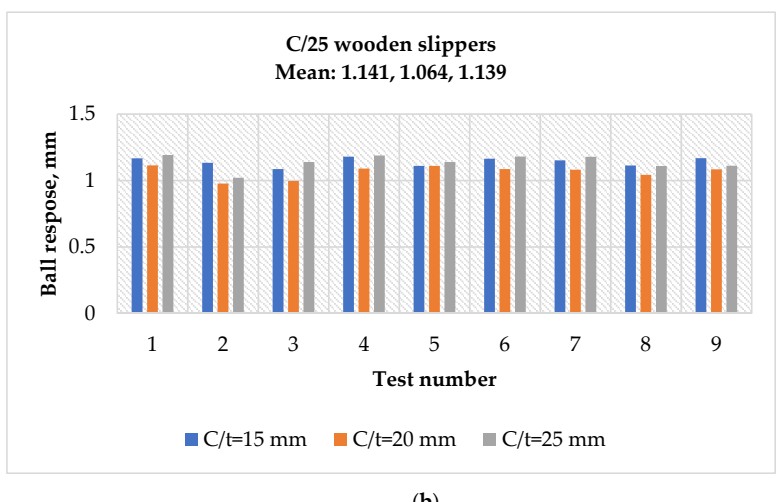

(**b**)

**Figure 11.** *Cont.*

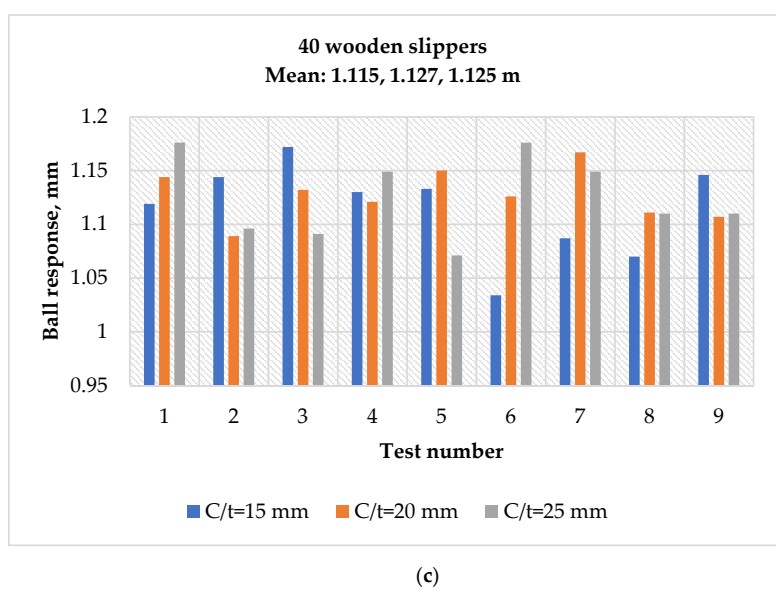

(**c**)

**Figure 11.** Ball response for C structure: (**a**) 13 wooden slippers; (**b**) 25 wooden slippers; (**c**) 40 wooden slippers.

It can be seen from Figure 10 that the thickest parquet, respectively with a thickness of 25 mm, has the best reaction of the ball on average of 1.174 m; this means that the rigid structures (in this case, the rigidity being given by the greater thickness of the frieze parquet) are more suitable for floor basketball gyms. All type B structures had a good ball reaction above the minimum acceptable value of 1.0836 m. The energy absorption of the three types of structures with values of 6.61, 6.00, and 1.88% was lower than the limiting energy value of 6.69%, which means that from this point of view, all type B structures are acceptable for use in the case of basketball halls. Similar results were observed by Acura et al. [18] by using eucalyptus wood, Same values were also sfound by Sepliarsky et al. [8], where oak wood was used.

Regarding the structures on the frame of softwood and beech wood shock pads, respectively the C-type ones, their analysis was made according to the number of taggers glued on the lower part of the structure, respectively with 13, 25, and 40 slippers. The results for each structure are presented in Figure 11.

First of all, Figure 11 shows a significant increase in the ball's response as the number of wooden slippers increases. Thus, for the thickness of the parquet frieze of 20 mm, an increase of the ball response was observed from 1.1086 to 1.115 mm (a percentage increase of 2.6%) when the number of wooden taggers increased from 13 to 40 pieces. Analyzing each structure separately, it was found that there were two structures with ball response below the allowable limit: (1) the structure C/t = 20 mm/13 wooden taggers with a value of 0.966 mm and an energy absorption of 13.2% and (2) the structure C/20 mm/25 taggers with a value of 1.064 mm and an energy absorption of 7.7%. All the other structures had a very good ball response and a corresponding energy absorption, which means that they can be used successfully to make the floors of basketball halls. Similar results were obtained by Turcas et al. [14,17] and others [10,18,27], which used beech parquet and similar structures.

The analysis of type D structures was done as in the case of type C structures, depending on the thickness of the parquet strip (15, 20, and 25 mm) and depending on the number of rubber slippers arranged on the lower face of the structure, respectively 13, 25, and 40 pieces.

From Figure 12, it is clearly observed that the increase in the slippers number led to the elasticity increase of the structure, and the ball's response decreased. For example, in the case of the 25 mm thick structure, the ball's response decreased from a value of 1.157 mm (for 13 rubber taggers) to 1.064 mm (for 40 rubber taggers), respectively a decrease of 7.9%. From this, it can be concluded that upon increasing the number of rubber taggers, the parquet structure will be more elastic, and the ball's response to touching the floor will be

smaller. The same value of the decrease was in the case of energy absorption. The smallest decreases in the ball's response were observed in the case of the number of rubber taggers of 40 pieces, in which case all three structures did not exceed the minimum threshold of 1.083 mm and are not accepted for use [17,27]. All the other six structures have exceeded the limiting values and thus are suitable for use for floors in basketball halls.

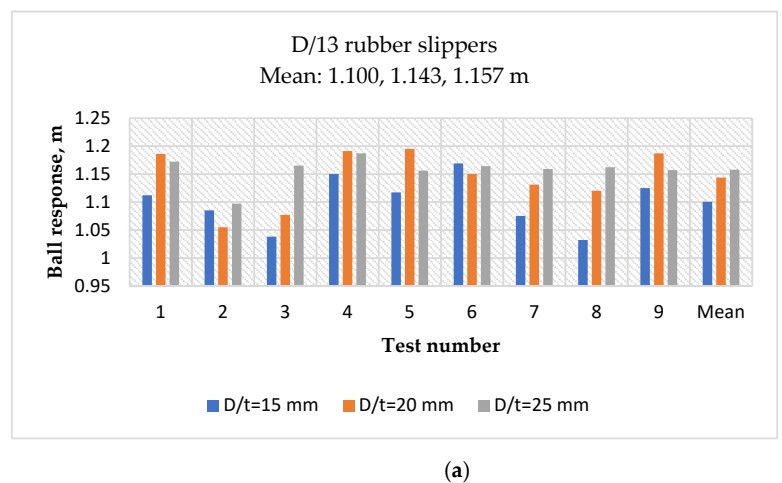

(**a**)

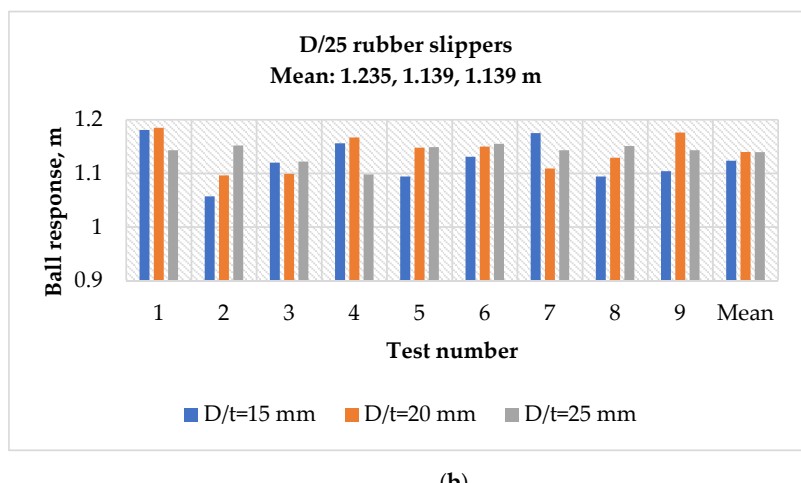

(**b**)

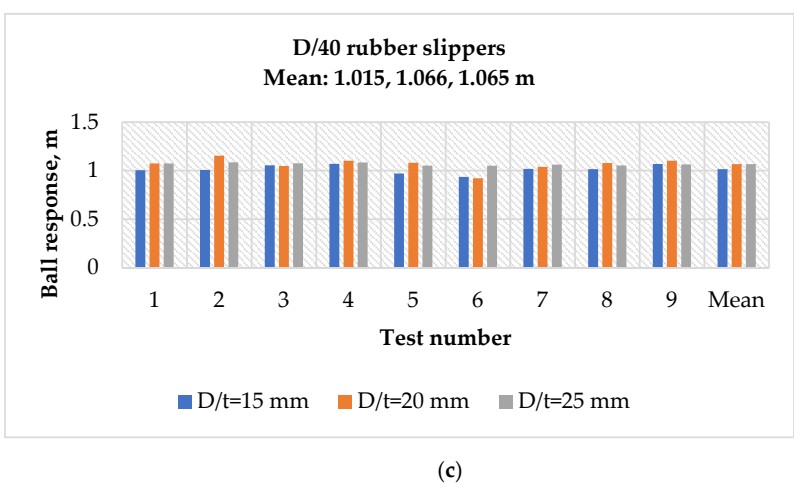

(**c**)

**Figure 12.** Ball response for D-type structure: (**a**) for structure with 13 rubber slippers; (**b**) for structure with 25 rubber slippers; (**c**) for structure with 40 rubber slippers.

## 4. Conclusions

Based on the experimental data but also based on the standards in the field, the minimum value of the ball's response on the parquet surface of 1.083 m was identified (representing 90% of the ball's response on the concrete surface) with a maximum energy absorption of 6.68%.

The structures with high rigidity and low elasticity had a better behavior of the basketball ball's response on the parquet surfaces.

The type A structures, which provided three and five longitudinal slats of softwood as support, did not correspond in terms of the basketball ball's response on the parquet surface, having values of 0.923 and 0.891 m, which were respectively 14.7 and 17.7% lower than the value limit of 1.083 m.

The type B structures, with the support of the parquet on a softwood frame provided with five stringers and five girders, had a very good ball response: all three structures exceeded the minimum reference value. This is due to the frame-type structure with five stringers and five girders joined in the welt, which ensures a good rigidity of the structure. However, it is recommended to use structures with a thickness of 20 and 25 mm due to the increased rigidity.

In the case of type C structures, with softwood frame and beech slippers, the increase in the number of taggers led to an increase in the rigidity of the structure. Therefore, the structure with 40 beech wood taggers with a basketball response of 1.115, 1.127, and 1.125 m is recommended, which were 2.8, 4.0, and 3.8% higher than the minimum reference value.

In the case of type D structures, which had a softwood frame support put on rubber slippers, with the increase of the number of rubber taggers at 25 and 40, there was an increase in the elasticity of the structure and a decrease in the ball's response to touching the floor under 1.083 m. That is why it is recommended to use structures with 13 rubber slippers, with energy absorption of 6.0, 3.6, and 2.8%, all of which are much lower than the maximum value of 6.68%.

**Author Contributions:** Conceptualization, A.L.; methodology, A.L.; software, A.L., L.R.; validation A.L., M.T.D., C.S. and L.R.; formal analysis, L.R.; investigation, A.L.; resources, M.T.D.; data curation L.R., writing—original draft preparation, C.S.; writing—review and editing, A.L.; visualization, A.L., supervision, A.L.; project administration, M.T.D.; funding acquisition, A.L. All authors have read and agreed to the published version of the manuscript.

**Funding:** This research received no external funding.

**Institutional Review Board Statement:** Not applicable.

**Informed Consent Statement:** Not applicable.

**Data Availability Statement:** Not applicable.

**Acknowledgments:** We would like to thank the Transilvania University of Brasov, for all the support provided in conducting the research and drafting the paper.

**Conflicts of Interest:** The authors declare no conflict of interest.

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
