# Peer review of "The Ball Response on the Beech Parquet Floors Used for Basketball Halls"

_applsci, doi:10.3390/app11177816_

Round 1

Reviewer 1 Report

The paper deals with the characterization of several parquet floors aimed to the manufacturing of basketball halls. The work is interesting and mostly well organized and written, but need few improvements to be clearer to the reader. My main comments are in the following:

The introduction presents several previous works on the matter, but more as a mere list. It should better focus on the research presented in the paper: the previous achievements should give the starting point of the research and justify the need of further investigations.

Material and methods, par 2.1: How many pieces were tested for each thesis (thickness / width / length)?

Par. 2.2: the structures tested were many with many different characteristics. A table summarizing their characteristics and differences would help the reader to follow the work

Figure 4: No letters were present in the figures, while in the text the authors refer to Fig.4a, b etc. Moreover, the line indicating the section A-A is not reported.

Figure 5, 9, 10 and 11: the unit is most likely wrong. It should be m and not mm.

Figure 6 is not described and discussed in the text.

Figure 5 and 6: please, take care to write captions that can be read without reading the text. What “force”?

Line 307: “for technological reasons”, please, be more specific

Figure 9: “sipci”. Again improve the caption.

Generally speaking, discussions are completely missing. The results are described but not discussed.

Author Response

Many thanks to reviewer 1 for their work. Here is our response, point by point:

  • In the Introduction chapter, some additions were made, in such a way as to demonstrate the need for the present research.
  • Total number of specimens for elasticity was added.
  • Table with all structure were added.
  • In Figure 4, different parts were numbered with a, b, c and d. Section A-A was explained.
  • In those figures the millimeter was changed by the meter, except for figure 5 where the millimetre remained because it was about the bending deformation and not the response of the ball.
  • Type of force was added.
  • A supplement has been added, for a better understanding of the text.
  • My native word "sipci" has been replaced with the word "strips"
  • More other discussions were added.

Authors

Reviewer 2 Report

I would like to appreciate the authors for their work in this paper. Suggestions are as follows:

  1. Overall the manuscript required clarity on scientific research findings.
  2. In conclusion, the statement mentioned requires more clarity: "That is why it is recommended to use structures with 13 rubber slippers, with an energy absorption of 6.0, 3.6 and 2.8% much lower than the maximum value of 6.68%."
  3. Further, authors can prefer more citations to justify the statements mentioned inthe manuscript. 

Author Response

Many thanks to reviewer 2 for the work done.

  1. More clarities were done in the manuscript, both in the methodology part and in the results part.
  2. In terms of conclusions, the paragraph in question has been completed in order to increase its clarity.
  3. The bibliographic citation was extended, especially in the part of results and discussions, in order to highlight the results of the research.

Authors